# Learning to ground decentralized multi-agent communication with contrastive learning

**Yat Long Lo & Biswa Sengupta**
Zebra Technologies, London, United Kingdom
{yat.longlo,biswa.sengupta}@zebra.com

## Abstract

For communication to happen successfully, a common language is required between agents to understand information communicated by one another. Inducing the emergence of a common language has been a difficult challenge to multi-agent learning systems. In this work, we introduce an alternative perspective to the communicative messages sent between agents, considering them as different incomplete views of the environment state. Based on this perspective, we propose a simple approach to induce the emergence of a common language by maximizing the mutual information between messages of a given trajectory in a self-supervised manner. By evaluating our method in communication-essential environments, we empirically show how our method leads to better learning performance and speed, and learns a more consistent common language than existing methods, without introducing additional learning parameters.

## 1 Introduction

Communication between agents is a key capability necessary for effective cooperation in multi-agent systems. To communicate successfully, a speaker and a listener must share a common language to have a shared understanding of the symbols being used (Dafoe et al., 2020). Developing algorithms to induce such a common language (i.e. *grounding communication*) for emergent communication has been a difficult open problem. Existing works have attempted to address this challenge based on three main directions, namely centralized learning, differentiable communication, and supervised learning. To begin with, Centralized learning approaches, like Foerster et al. (2016) and Lowe et al. (2017), share models among agents which implicitly ground communication to a common language to a certain extent. Differentiable communication approaches including Foerster et al. (2016) and Sukhbaatar et al. (2016) offer another grounding feedback by allowing gradients to flow through across agents. Lastly, supervised learning like Graesser et al. (2019) leverages ground-truth information to adjust messages to obtain greater task rewards.

However, none of these existing methods has a resemblance to how communication emerges in nature. Specifically, considering how human languages emerge, no central modules or communication-specific supervised signals were needed (Nowak & Krakauer, 1999). In other words, beings in nature learn to communicate resembles more a fully decentralized setting. Yet, existing decentralized approaches to communication in multi-agent reinforcement learning (MARL) systems are known to perform poorly even in simple tasks (Foerster et al., 2016). The main challenge lies in a lack of common grounding in communication, making it difficult for agents to communicate meaningfully (Lin et al., 2021). Recent work like Eccles et al. (2019) and Lin et al. (2021) propose novel methods to ground communication in this setting. The former introduces biases to the loss function encourage positive listening and positive signaling, which was subsequently shown to be ineffective in sensory-rich observations (Lin et al., 2021). (Lin et al., 2021) propose using autoencoder to ground communicative messages by having each agent reconstruct their observations. However, existing works have not considered the relationships among messages sent by different agents as a direction for grounding, which is arguably essential given how we would like to have all the agents produce messages under a common language.

In this work, we introduce an alternative perspective based on the relationship between messages an agent produces and the messages it receives. This leads to a simple method based on contrastive learning to ground communication. Precisely, inspired by literature in representation learning across different views of a data sample (Bachman et al., 2019), for a given trajectory, we propose viewing messages sent across agents to be different incomplete views of environment states. From this perspective, messages in a trajectory should be more coherently constructed than messages sent in another trajectory. Hence, we propose using a contrastive learning method to align the message space for communication (i.e. *ground communication*) which pulls messages within a trajectory to be closer to each other and pushes messages of different trajectories to be further apart. We evaluate our method in communication-essential environments and empirically show how our method leads to improved speed and performance with a greater resemblance of a common language than existing methods, without additional learning parameters.

## 2 PRELIMINARIES

We base our investigations on decentralized partially observable Markov decision processes (Dec-POMDPs) with $N$ agents to describe a *fully cooperative multi-agent task* (Oliehoek & Amato, 2016). A Dec-POMDP consists of a tuple $G = \langle S, U, P, R, Z, \Omega, n, \gamma \rangle$. $s \in S$ is the true state of the environment. At each time step, each agent $i \in N$ chooses an action $a \in A^i$ to form a joint action $\boldsymbol{a} \in \boldsymbol{A} \equiv A^1 \times A^2 ... \times A^N$. It leads to an environment transition according to the transition function $P(s'|s, a^1, ...a^N) : S \times \boldsymbol{A} \times S \rightarrow [0, 1]$. All agents share the same reward function $R(s, \boldsymbol{a}) : S \times \boldsymbol{A} \rightarrow \mathbb{R}$. $\gamma \in [0, 1)$ is a discount factor. As the environment is partially observable, each agent $i$ receives individual observations $z \in Z$ based on the observation function $\Omega^i(s) : S \rightarrow Z$.

We denote the environment trajectory and the action-observation history (AOH) of an agent $i$ as $\tau_t = s_0, \mathbf{a_0}, ....s_t, \mathbf{a_t}$ and $\tau_t^i = \Omega^i(s_0), a_0^i, ....\Omega^i(s_t), a_t^i \in T \equiv (Z \times A)^*$ respectively. A stochastic policy $\pi(a^i|\tau^i) : T \times \boldsymbol{A} \rightarrow [0, 1]$ conditions on AOH. The joint policy $\pi$ has a corresponding action-value function $Q^\pi(s_t, \boldsymbol{a_t}) = \mathbb{E}_{s_{t+1:\infty}, \boldsymbol{a_{t+1:\infty}}}[R_t|s_t, \boldsymbol{a_t}]$, where $R_t = \sum_{i=0}^{\infty} \gamma^i r_{t+i}$ is the discounted return. $r_{t+i}$ is the reward obtained at time $t + 1$ from the reward function $R$.

To account for communication, similar to Lin et al. (2021), at each time step $t$, an agent $i$ takes an action $a_t^i$ and produces a message $m_t^i = \Psi^i(\Omega^i(s_t))$ after receiving its observation $\Omega^i(s_t)$ and messages sent at the previous time step $m_{t-1}^{-1}$, where $\Psi^i$ is agent $i$'s function to produce a message given its observation and $m_{t-1}^{-1}$ refers to messages sent by agents other than agent $i$. The messages are assumed to be vectors of either discrete or continuous values. Here, we use continuous messages.

## 3 METHODOLOGY

We propose a different perspective on the message space used for communication. At each time step $t$ for a given trajectory $\tau$, a message $m_t^i$ of an agent $i$ can be viewed as an incomplete view of the environment state $s_t$ because it is a function of the environment state as formulated in section 2. Naturally, messages of all the agents $\mathbf{a_t}$ are different incomplete perspectives of $s_t$. To ground decentralized communication, we hypothesize that we could leverage this relationship between messages within a trajectory to encourage consistency and proximity of the messages across agents. Specifically, we propose maximizing the mutual information between messages within a trajectory using contrastive learning which aligns the message space by pushing messages of the same trajectory closer together and messages of different trajectories further apart.

We extend the recent supervised contrastive learning method (Khosla et al., 2020) to the MARL setting by considering multiple trajectories during learning. We refer to this loss formulation as *Communication Alignment Contrastive Learning (CACL)*. In this case, we consider messages within a trajectory to be different views of the same data sample with the same label. Let $i \in I$ be an index of a message in a batch. $j$ be an index of a trajectory in a batch of trajectories $H$, $\mathbf{m_j}$ be a set of messages sent in a trajectory $j$ and $A(i) \equiv I \setminus \{i\}$. The loss is in the form of:

$$L_{CACL} = \sum_{i \in I} \frac{-1}{|P(i)|} \sum_{p \in P(i)} \log[\exp(m_i \cdot m_p/\tau)] - \log[\sum_{a \in A(i)} \exp(m_i \cdot m_a/\tau)] \tag{1}$$

Here, $P(i) \equiv \{p \in A(i) : m_p \in \mathbf{m_j}\}$ is the set of messages within a trajectory which are viewed as positive examples distinct from $i$. $|P(i)|$ is its cardinality and $\tau \in \mathbb{R}^+$ is a scalar temperature.

Practically, each agent has a replay buffer that maintains a batch of trajectory data containing messages received during training to compute the *CACL* loss. Similar to Khosla et al. (2020), messages are normalized before the loss computation and a low temperature (i.e. $\tau = 0.1$) is used which empirically shows benefits in performance. Together with the reinforcement learning (RL) loss $L_{RL}$, the total loss is formulated as follows:

$$L = L_{RL} + \kappa L_{CACL} \tag{2}$$

where $\kappa$ is a hyperparameter to scale the *CACL* loss.

## 4 EXPERIMENTS AND RESULTS

### 4.1 EXPERIMENTAL SETUP

We evaluate our method on two communication-essential environments. These environments require meaningful communication to improve task performance given the limited information each agent has.

**Predator-Prey**: Agents (i.e. predators) have the cooperative goal to capture a moving prey, by having more than one predator approaching the prey. To make it communication-essential, we devise a variant called *Fully-Cooperative Predator-Prey*, with 4 predators and 2 preys. Agents are required to entirely surround a prey for it to be captured while they cannot see each other in their fields of view. Therefore, communication is important for agents to communicate their positions and actions. We evaluate each algorithm with episodic rewards during evaluation episodes.

**Traffic-Junction**: Proposed by Sukhbaatar et al. (2016), it consists of A 4-way traffic junction with cars entering and leaving the grid. The goal is to avoid collision when crossing the junction. We use 5 agents with a vision of 1. We evaluate each algorithm with success rate during evaluation episodes.

Details of the environment parameters can be found in appendix A.1. All results are averaged over 12 evaluation episodes over 6 random seeds.

### 4.2 TRAINING DETAILS

For baselines, we compare our methods against *AE COMM* (Lin et al., 2021) which grounds communication by reconstructing encoded observations, DIAL (Foerster et al., 2016) which learns to communicate through differentiable communication, and independent RL without communication.

All methods use the same architecture based on the independent actor-critic algorithm with n-step returns and asynchronous environments (Mnih et al., 2016). Each agent has an observation and message encoder to process observations and received messages. For methods with communication, each agent has a communication head to produce a message based on encoded observations. For policy learning, a GRU (Dey & Salem, 2017) is used for partial observability before the policy and value heads. Each head is a 3-layer fully-connected neural network. We perform spectral normalization in the penultimate layer for each head to improve training stability (Gogianu et al., 2021). Details for architecture and hyperparameters used can be found in appendix A.2

### 4.3 RESULTS

Figure 4.3 shows the performances of our proposed method and the baseline methods. Our proposed method *CACL* outperforms the baseline methods in terms of both final performance and learning speed. Notably, despite not having extra learning parameters, it outperforms *AE COMM* (Lin et al., 2021) which grounds communication by reconstructing encoded observations with an additional decoder per agent.

To investigate the potential reasons behind our method's improvement over the baselines, we look at the agents' messages sent during evaluation episodes in the Fully-Cooperative Predator-Environment and compare our method against the most performant baseline *AE COMM*. Specifically, we collect

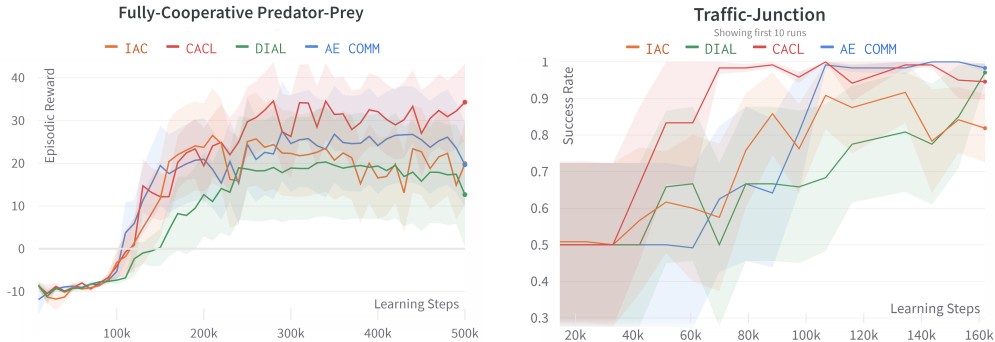

Figure 1: Comparing the performance of our method with baseline methods. Left: Our method *CACL* is able to achieve better episodic reward than the baselines in the Fully-Cooperative Predator-Prey environment without additional learning parameters. Right: Our method *CACL* is able to achieve a better success rate than the baselines in the Traffic-Junction environment with greater learning speed. Standard errors are plotted as shaded areas.

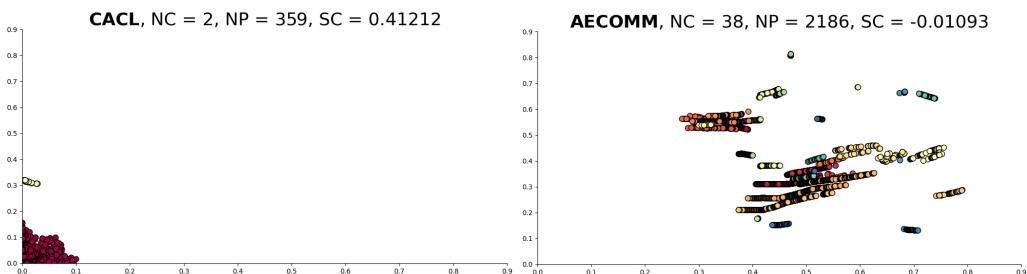

Figure 2: Clustering of messages over evaluation episodes on our method *CACL* and the most performant baseline *AE COMM*. NC, NP ,and SC refer to the number of clusters identified, number of noise points, and silhouette coefficient. Our method empirically appears to form a more consistent coherent language with significantly better SC.

all the messages sent from 7 evaluation episodes and perform clustering on them using DBSCAN (Ester et al., 1996). Figure 4.3 shows the clustering results for both methods. Our proposed method appears to be able to learn a more coherent and consistent common language as the clustering algorithm detects a smaller number of clusters and noise points with a much higher silhouette coefficient than *AE COMM*. Silhouette coefficient has a value between -1 and 1 which measures how well defined the clusters are. The higher the value the less overlapping between clusters. One plausible explanation to these results is that *AE COMM* grounds communication without considering messages of other agents, leading to the formation of multiple protocols between agents within the same message space, while our proposed method *CACL* induces a more common protocol with the consideration of messages received. The improved performance and speed of our approach further indicate the benefits of learning a common language in the decentralized communication setting.

## 5 CONCLUSION AND FUTURE WORK

In this work, we introduce an alternative perspective to ground communication in the decentralized MARL setting by considering the relationship between messages sent and received within a trajectory. Using this perspective, we propose a method to ground communication without additional learning parameters based on contrastive learning. We experimentally show how our proposed method leads to better performance and learning speed by learning a more coherent and consistent common language among agents. For future work, we aim to explore our method's effectiveness in environments with many agents and extensions to ground communication with better interpretability.

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

# A    APPENDIX

## A.1    ENVIRONMENT DETAILS

Figure A.1 provides a visual illustration of the environments used.

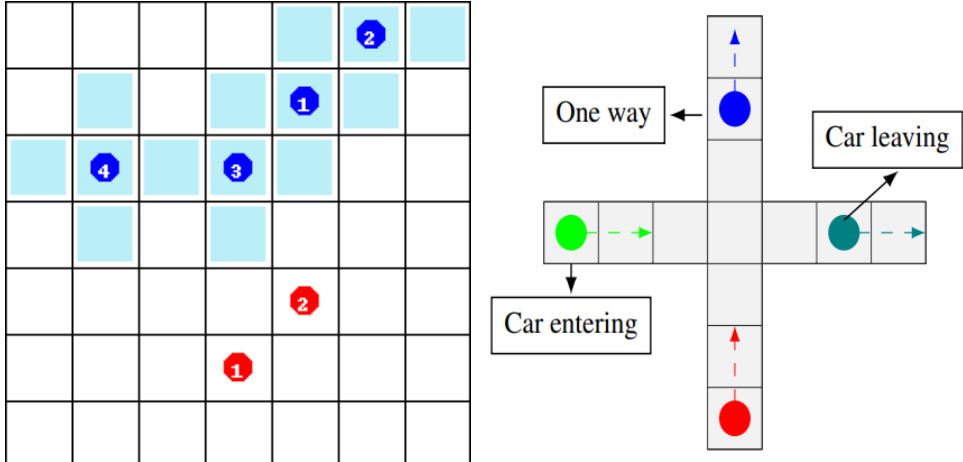

Figure 3: Visual illustration of the environments used. Left: Fully-Cooperative Predator-Prey, taken from Koul (2019). Right: Traffic-Junction, taken from Singh et al. (2018)

### A.1.1    FULLY-COOPERATIVE PREDATOR-PREY

We modify the Predator-Prey implementation by Koul (2019). Fully-Cooperative Predator-Prey has a higher communication and coordination requirement than the original Predator-Prey environment. Specifically, for a prey to be captured, it has to be entirely surrounded (i.e. the prey cannot move to another grid position in any actions).

Here, we use and 7x7 gridworld. In each agent's observation, it can only see the prey if it is within the field of view (3x3) and cannot see where other agents are. A shared reward of 10 is given for a successful capture and A penalty of -0.5 is given for a failed attempt. A -0.01 step penalty is also applied per step. Each agent has the actions of *LEFT*, *RIGHT*, *UP*. *DOWN* and *NO-OP*. The prey has the movement probability vector of $[0.175, 0.175, 0.175, 0.175, 0.3]$ with each value corresponding to the probability of each action taken.

All algorithms are trained for 30 million environment steps with a maximum of 200 steps per episode.

### A.1.2    TRAFFIC-JUNCTION

We use the Traffic-Junction environment implementation provided by Singh et al. (2018). The gridworld is 8x8 with 1 traffic junction. The rate of cars being added has a minimum and maximum of 0.1 and 0.3. We use the easy version with two arrival points and 5 agents. Agents are heavily penalized if a collision happens and have only two actions, namely *gas* and *brake*.

All algorithms are trained for 10 million environment steps with a maximum of 20 steps per episode.

## A.2    ARCHITECTURE AND HYPERPARAMETERS

Figure 4 illustrates the components of the architecture used in this work, similar to (Lin et al., 2021). A message head is only used for algorithms with communication, namely *CACL*, *AE COMM* and *DIAL*. The Grounding Module refers to mechanisms to ground the messages produced by the message head, used in *CACL* and *AE COMM*. Unless specified otherwise, we fix all hidden layers to be a size of 32. The observation encoder and message encoder output values of size 32 and 16 respectively. Messages received are concatenated before passing to message encoders. For all

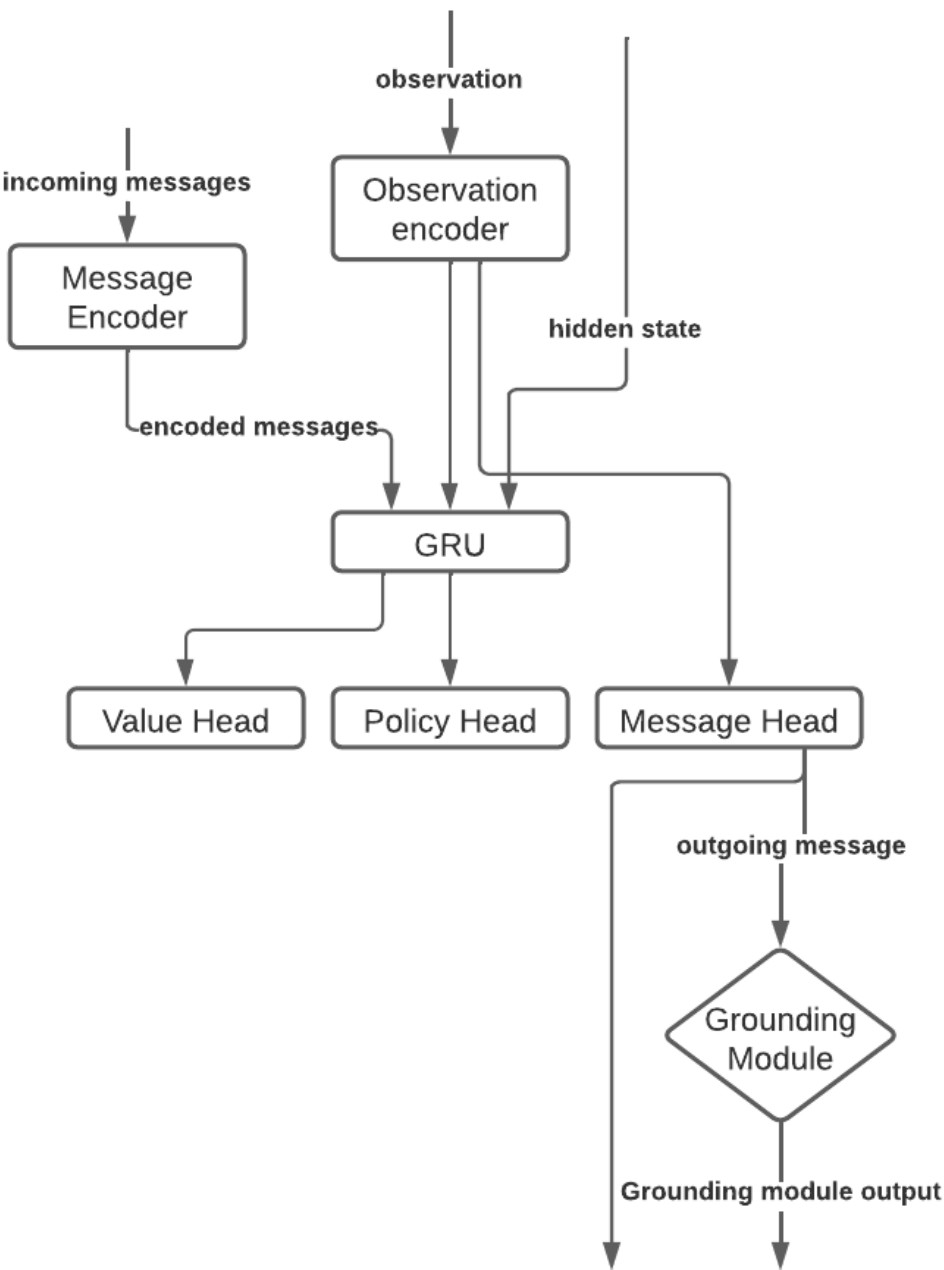

Figure 4: Architectural illustration for algorithms with communication. To remove communication, the message head is disabled. Grounding module is only relevant to *CACL* and *AE COMM*. The former is a loss function and the latter is a decoder to reconstruct the encoded observation.

the methods with communication, they produce messages of length 4 with a sigmoid function as activation. All models are trained with the Adam optimizer (Kingma & Ba, 2014).

Table 1 lists out the hyperparameters used for all the methods.

| Learning Rate | 0.0003 |
|---|---|
| Epsilon for Adam Optimizer | 0.001 |
| $\gamma$ | 0.99 |
| Entropy Coefficient | 0.01 |
| Value Loss Coefficient | 0.5 |
| Gradient Clipping | 2500 |
| $\tau$ for *CACL* | 0.1 |
| $\kappa$ for *CACL* | 0.5 |
| Number of Asynchronous Processes | 12 |
| N-step Returns | 5 |

Table 1: Table for hyperparameters used across methods

