# OpenReview forum: "Learning To Ground Decentralized Multi-Agent Communication with Contrastive Learning"
_ICLR.cc/2022/Workshop/EmeCom — EmeCom Workshop at ICLR 2022_

### Official Review · Reviewer_Cuyr · 2022-03-21

**Rating:** Strong Accept
**Confidence:** 5

**Review:**

The paper frames this original idea: In a embodied agent, intermediate messages are a partial view of the state. From a self-supervised perspective, those can be seen as view. Therefore, they are suitable transformation to apply an InfoNCE loss.
It is simple, somehow a bit (positively) naive, and quite refreshing and an original idea.

I would encourage the authors to ease the readability of the InfoNCE loss, and the formalism as it is a bit entanbgled. Although familiar with SSL methods, it took me multiple reads to correctly link the equation's terms together.

I would also cite a few papers that leverage SSL idea to enhance emergent communication:
 - https://proceedings.neurips.cc/paper/2021/file/e250c59336b505ed411d455abaa30b4d-Paper.pdf
 - https://openreview.net/forum?id=AUGBfDIV9rL
 - https://arxiv.org/abs/2109.09390

The experimental setting may be far too toyish, but it is an excellent first proof of concepts. I would encourage the authors to look at more complex environment to explore it (color maze, maze solving, overcook, or even Talk the walk).

I ackowledge the quality of the appendix and experimental protocol details.. 6 random seeds is a nice (although still not enough...) good practice.

In the end, I strongly recommend paper acceptance for it is exactly what we can expect from a workshop paper. Good work!



**Best Paper Award:**

Yes

---

### Official Review · Reviewer_iFGC · 2022-03-21

**Rating:** Accept
**Confidence:** 5

**Review:**

Summary of the contributions:

In the context of an episodic, communication-enabled, fully-collaborative, embodied, multi-agent setting, this paper tackles the problem of inducing the emergence of a common language for the considered population of agents.
Following the intuition that messages exchanged during one trajectory/episode could be treated as different viewpoints of a common context, and that, in a contrapositive fashion, the messages sourced from different trajectories/episodes
should not have anything in common, the paper proposes an auxiliary loss inspired by contrastive learning. This contrastive loss aims to maximize the mutual information (MI) between messages sourced from a given trajectory/episode.

The current paper chooses to work with continuous messages only. The paper evaluates the resulting agent on two relevant benchmarks (Predator-Prey and Traffic-Junction) against three baselines (Independent Actor Critic (IAC), Differentiable Inter-Agent Learning (DIAL), and grounding multi-agent communication through autoencoding (AE-COMM)), comparing against the following metrics: final performance and sample efficiency.

Finally, the (continuously-valued) emerging protocols/languages are investigated in terms of clustering in the message landscape.


Novelty, relevance, significance:

1) The idea of the paper is original and thoroughly executed upon.

2) The paper leverages (self-supervised) contrastive learning in the context of emerging communication, it is therefore very relevant and topical to the community of the workshop.

3) Given the context of this workshop, the paper is more than significant. Nevertheless, in order to provide feedback towards the writing of a conference paper, I think that the paper overlooks aspects that would see the proposed auxiliary loss as a positive signaling induction bias (Lowe et al. [2019], Eccles et al. [2019]), I will touch upon this further below. But, as a result of this subsequent aspect, I would expect a similar agent than that of [Eccles et al., 2020] to be part of the baseline lineup for this paper to be considered complete and significant to
the overall community.


Soundness:

The paper claims that the proposed algorithm (i) outperforms the baseline in terms of final performance, (ii) has greater sample efficiency than baselines, (iii) induces a ”more coherent and consistent common language” than the AE-COMM baseline would, and (iv) achieves all of the above without adding any hyperparameters.

Claims (i) and (ii) are substantiated by Figure 1 (left and right). While the number of random seeds (6) is good enough, I would advise for a larger sample. Moreover, I do not think that the learning curves presented in Figure 1 sufficiently substantiate the claims: the final performance distributions seem to overlap strongly from one algorithm to another, especially on the Predator-Prey benchmark, and the sample efficiency of an algorithm is better reported upon via a measure of the Area-Under-the-Curve (AUC).

Thus, I would advise the authors to:
•present the final performance of the different algorithms in table with mean and std deviation/error values, and perform a statistical test showing statistical significance of their results (e.g. using two-sample Kolmogorov-Smirnov test : https://docs.scipy.org/doc/scipy/reference/generated/scipy.stats.ks 2samp.html), and
•present a measure of AUC in a table (mean +/- std) and again perform a statistical test.

Claim (iii) seems fairly substantiated by Figure 2. Nevertheless, it might be interesting to show a before/after convergence of the CACL-specific loss to properly distill in the paper the qualitative effects it has on the messages clusters. I would propose to the authors to try to show the CACL-specific loss curve with some anchor points where the corresponding clustering method results would be shown.
Minor: Which benchmark was the clustering analysis performed on? I am hoping it was on TrafficJunction where 2 meanings are indeed sufficient, if I understand correctly (BRAKING or ADVANCING). Thus, it could be interesting to qualitatively investigate the context in which some messages out of each clusters are send, for instance by presenting some of the state observations next to the corresponding message’s cluster ID.

Claim (iv) is problematic in many ways, as far as I understand:
- the addition of this auxiliary loss naturally creates an extra hyperparameter in the form of the loss weighting coefficient,
 - the CACL loss function contains a temperature hyperparameter τ.

Firstly, both hyperparameters may be environment-dependent values (until shown otherwise by an hyperparameter search, maybe?) so as presented the paper does not substantiate claim (iv) or its weaker form: “there is no hyperparameter to tune”.
I would advise the author to reformulate this claim towards something along the line of: “the proposed algorithm is robust to variations of the hyperparameters it involves.”
Subsequently, while testing on two benchmarks is commandable, I doubt it is enough to claim hyperparameter robutstness of the approach, and would thus advise for explicit inclusion of some other benchmarks or for, at least, implicitly increasing the number of context in which the algorithm is tested by varying the hyperparameters of the current benchmarks (e.g. grid size, visibility size, parameters of the communication protocol...).
Thus, in order to substantiate the reformulated claim (iv), I would welcome a thorough analysis showing that the final performance and AUC distributions (from one set of hyperparameter to another) are all statistically significantly similar.


Quality of writing/presentation & Literature:

1) The paper is well-organised and and the attention to details in explaining training/implementation setup is very well appreciated.

2) In terms of the literature, I think the positioning of the paper within the literature may need further clarification:

Regarding the categorization of the different directions (centralized/decentralized, differentiable communication protocol or not, and supervised (SL) vs selfplay/MARL), firstly, I would argue that ?is leveraging both SL and RL, but SL is not leveraged at the level of learning a common protocol. Thus, I am not sure this distinction makes sense, or else I may be misunderstanding the point that the authors were trying to make so I would advise for a reformulation.

Second, and most importantly, I think this paper investigates one way to bias for positive signaling, and it is in that nuance that the contribution shines the most. Indeed, Eccles et al. [2019] and Lin et al. [2021] both investigate different positive signaling induction biases, towit and respectively, either in the sense of furthering high MI between a speaker agent’s trajectory and its messages, or in the sense of state-communication autoencoding. Thus, this paper investigates yet another way of biasing for positive signaling, this time in the sense of furthering high MI between the messages of all speaker agents within a given trajectory.

Given the importance of the concepts of positive signaling (PS) and positive listening, I strongly believe that this paper is at its most relevant and significant when positioned within that context.

Minor changes:
- the figure for the clustering is mislabelled in the text. Figure 4.3 instead of Figure 2.
- in the legend of Figure 1, IAC is used to refer to Independant Actor Critic, I presume, but the paper did not specify it (maybe worth adding in the first paragraph of Section 4.2 Training Details).


References:

T. Eccles, Y. Bachrach, G. Lever, A. Lazaridou, and T. Graepel. Biases for emergent communication in multi-agent reinforcement learning. Dec. 2019.

T. Lin, M. Huh, C. Stauffer, S.-N. Lim, and P. Isola. Learning to ground Multi-Agent communication with autoencoders. Oct. 2021.

R. Lowe, J. Foerster, Y.-L. Boureau, J. Pineau, and Y. Dauphin. On the Pitfalls of Measuring Emergent Communication. mar 2019. URL http://arxiv.org/abs/1903.05168.

---

### Comment · Program_Chairs · 2022-04-01
**Runner-up Best Paper**

After going through all accepted papers, the program chairs have decided that this work is one of the best 3 papers at the workshop and a runner-up for the best paper award. We would like to congratulate the authors!

We found the connection between contrastive learning views and emergent communication messages to be incredibly insightful. Then using that connection to improve grounding and connect the fields of self-supervised learning and emergent communication is an interesting, novel direction. The experiments and qualitative evaluations are interesting and demonstrate a good testing ground. This work encapsulates exactly the type of "New Frontiers" we were aiming for. Congratulations!

---

### Decision · Program_Chairs · 2022-03-25

**Decision:**

Accept

**Comment:**

Both reviewers were impressed with this paper and we believe it would make for excellent discussion at the workshop. We congratulate the authors and look forward to discussing this at the workshop. We hope they can use the feedback from reviews to make a strong conference submission